# Hemopexin Suppresses Hepatocellular Carcinoma via TNF-α-Mediated Mitochondrial Apoptosis

**DOI:** 10.3390/cancers17182969

**Published:** 2025-09-11

**Authors:** Liying Ren, Yuxin Man, Xue Zhang, Qian Guo, Shaoping She, Yao Yang, Ran Fei, Xu Cong, Dongbo Chen, Wen Wei, Hongsong Chen

**Affiliations:** 1Peking University Hepatology Institute, Peking University People’s Hospital, Infectious Disease and Hepatology Center of Peking University People’s Hospital, Beijing Key Laboratory of Hepatitis C and Immunotherapy for Liver Diseases, Beijing International Cooperation Base for Science and Technology on NAFLD Diagnosis, Beijing 100044, China; renliying@bjmu.edu.cn (L.R.); zhangxue0205@bjmu.edu.cn (X.Z.); sheshaoping@pku.edu.cn (S.S.); yangyao927@bjmu.edu.cn (Y.Y.); 02659@pkuph.edu.cn (R.F.); congxu@pkuph.edu.cn (X.C.); 2Department of Medical Oncology, Sichuan Clinical Research Center for Cancer, Sichuan Cancer Hospital & Institute, Sichuan Cancer Center, University of Electronic Science and Technology of China, Chengdu 610041, China; manyuxin@scszlyy.org.cn (Y.M.); guoqian@scszlyy.org.cn (Q.G.); 3Peking University Third Hospital, Beijing 100191, China

**Keywords:** fibrinolysis, hepatocellular carcinoma, hemopexin, apoptosis

## Abstract

Hepatocellular carcinoma (HCC) remains a leading cause of cancer death, and better biomarkers and therapeutic targets are needed. We investigated the characteristics of fibrinolysis—an essential process that balances blood clot breakdown—and found that they divide HCC into biological subtypes with different tumor microenvironments and outcomes. Among these genes, hemopexin (*HPX*), a secreted protein that scavenges free heme, consistently showed a tumor-suppressive pattern in HCC. We built and externally validated a six-gene prognostic signature including *HPX* that stratifies patients by risk. Functional assays in multiple HCC cell lines and mouse models showed that increasing *HPX* levels reduced tumor growth and promoted programmed cell death through mitochondrial pathways linked to tumor necrosis factor-α. These results suggest that *HPX* may have value for risk assessment and could inform future strategies to modulate the tumor microenvironment in HCC.

## 1. Introduction

Cancer remains one of the leading causes of mortality worldwide, with an estimated 20 million new cases and 9.7 million deaths in 2022 [1]. The incidence is increasing in many developing countries due to population aging, lifestyle changes, and environmental factors, although trends vary across cancer types and regions [2]. Hepatocellular carcinoma (HCC) most commonly develops on a background of chronic liver disease. Major causes include chronic infection with hepatitis B virus (HBV) or hepatitis C virus (HCV) and long-standing hepatic fibrosis or cirrhosis arising from alcohol-related liver disease or non-alcoholic steatohepatitis (NASH) [3]. Despite continuous advances in cancer diagnosis and treatment, outcomes for HCC remain unsatisfactory, underscoring the need for mechanistically grounded biomarkers and therapeutic targets [4].

Fibrinolysis, the process of fibrin clot degradation mediated by the plasminogen–plasmin system, plays crucial roles in both physiological and pathological processes. Basically, fibrinogen is cleaved by thrombin to release fibrinopeptides A and B (FpA/FpB), exposing polymerization sites that drive fibrin assembly, and then factor XIIIa cross-links fibrin and stabilizes the clot. Fibrinolysis counterbalances this process: tissue-type or urokinase-type plasminogen activator converts plasminogen to plasmin, which degrades cross-linked fibrin into D-dimer and related fragments [5]. Under normal conditions, fibrinolysis ensures vascular patency by preventing excessive clot formation. However, dysregulation of this system contributes to various diseases, including cancer [6]. Increasing evidence highlights the significant role of fibrinolysis in tumor biology, where it influences tumor progression, metastasis, and angiogenesis through diverse mechanisms.

In gastric cancer (GC), key fibrinolytic and coagulation-related proteins, such as tissue factor (TF), thrombin, fibrinogen (FBG), FpA, and D-dimer (DD), are abnormally expressed and serve as biomarkers for disease progression and prognosis [7]. TF overexpression is associated with advanced GC and vascular invasion [8,9], while thrombin dysregulation reflects early coagulation changes [10]. Elevated fibrinogen levels correlate with poor prognosis and metastasis [11,12,13,14], increased FpA is associated with disease progression, including lymph node metastasis [15,16], and DD reflects fibrinolysis activation, indicating tumor spread and poor outcomes [17,18]. Similarly, in breast cancer, elevated levels of urokinase-type plasminogen activator (uPA), its receptor (uPAR), and plasminogen activator inhibitor-1 (PAI-1) are associated with tumor aggressiveness and poor prognosis [19]. The uPA/uPAR complex promotes metastasis by facilitating extracellular matrix degradation, cell migration, and angiogenesis through interactions with vitronectin, integrins, and EGFR, activating pathways such as RAF–MEK–ERK and VEGF [20,21].

On this basis, this study aims to investigate the role of fibrinolysis in HCC. We then develop and validate a prognostic signature derived from fibrinolysis-related characteristics and examine its clinical relevance and prognostic value. Finally, we nominate a priority candidate gene for functional evaluation and explore pathways that may underlie its effects.

## 2. Methods

### 2.1. Data Acquisition and Processing

TCGA pan-cancer gene expression data, copy number variation (CNV) data, and clinical information were obtained from UCSC Xena (https://xenabrowser.net/, accessed on 22 May 2023). The protein-coding transcriptome data, clinical information of HCC samples, and the protein-coding transcriptome data of liver tissue samples were downloaded from the TCGA database (TCGA-LIHC, https://portal.gdc.cancer.gov/projects/TCGA-LIHC, accessed on 9 June 2023). LIRI-JP data were obtained from the ICGC database (https://dcc.icgc.org/releases, accessed on 28 May 2023). GSE54236 data were obtained from the GEO database (https://www.ncbi.nlm.nih.gov/gds, accessed on 22 May 2023) [22]. RNA-seq data were processed using the DEseq2 package (v1.34.0) [23], and microarray data were processed using the limma package (v3.50.3) [24]. The tumor stemness data were obtained from Malta TM et al. [25].

### 2.2. Gene Set Function Enrichment Analysis and Tumor Microenvironment Analysis

The fibrinolysis gene set was obtained from the Gene Ontology (GO) database (www.geneontology.org) [26] and extracted from the MSigDB database “c5.go.v2024.1.Hs.symbols.gmt“ (accessed on 14 February 2025) [27]. Cell-death-related gene sets were sourced from Sun et al. [28]. “h.all.v2024.1.Hs.symbols.gmt” was downloaded from MSigDB database (https://www.gsea-msigdb.org/gsea/msigdb, accessed on 8 January 2025) [27]. Gene set enrichment analysis was performed using the “clusterProfiler” package (v4.12.2) [29]. Gene set function enrichment was scored through the GSVA package (v1.42.0). The degree of immune cell infiltration in the tumor microenvironment was predicted by using xCell (v1.1.0) [30]. The tumor microenvironment was scored using the “ESTIMATE” package (v1.0.13) [31]. The receptor–ligand pairs related to the tumor microenvironment were analyzed using the IOBR package (v0.99.0) [32].

### 2.3. Molecular Subtyping of Fibrinolysis in HCC and Weighted Gene Co-Expression Network Analysis

Molecular typing of HCC samples was performed using the fibrinolysis gene set by non-negative matrix factorization (NMF) (v0.26). The optimal number of subtypes of fibrinolysis-related molecular subtypes in HCC was determined based on the cophenetic correlation. We included the fibrinolysis molecular subtypes of HCC in the weighted gene co-expression network analysis (WGCNA) (v1.72-1) [33]. The optimal soft threshold β that met the requirements of the scale-free network was determined, and the co-expression network was constructed by the one-step method. The gene module significantly related to the fibrinolysis molecular subtypes was selected. The hub genes in the modules were screened based on a gene significance (GS) greater than 0.4 and module membership (MM) greater than 0.5 for further study.

### 2.4. Construction of the Risk Signature

Univariate Cox analysis was performed on the genes related to the fibrinolysis-related molecular subtypes in HCC. Genes with *p* < 0.05 were included in the least absolute contraction and selection operator (LASSO) analysis for further-related gene screening. Finally, a risk signature related to fibrinolysis molecular subtypes in HCC was then constructed through the step method of multivariate Cox regression analysis. The formula was established as follows:(1)$$RiskScore\;=\;{h_{0}\left(t\right)\exp\sum }_{j=1}^{n}Coefj\times Xj$$

The predictive performance of the risk signature was quantified through the area under the receiver operating characteristic curve (AUC).

### 2.5. Animals and Xenograft Tumor Model

Male BALB/c nude mice (4–6 weeks old) were purchased from Beijing Vital River Laboratory Animal Technology Co., Ltd (Beijing, China). Mice were housed under specific pathogen-free conditions in groups of 5 per cage. All procedures were approved by the Institutional Animal Care and Use Committee of Peking University People’s Hospital (approval number 2020PHE057) and conducted in accordance with ethical guidelines. For tumor implantation, 5 × 10^6^ cells suspended in 120 μL phosphate-buffered saline (PBS, Solarbio, Beijing, China) were subcutaneously injected into the flank of each mouse (each group: *n* = 5). Tumor growth was monitored every three days after palpable nodules appeared by measuring the longest (L) and shortest (S) diameters using calipers. Tumor volume was calculated as *V* = *L* × *S*^2^ × 0.5. Mice were randomized into experimental groups prior to tumor inoculation. Investigators performing measurements were blinded to group allocation to reduce bias. On day 19 post-inoculation, mice were euthanized humanely by cervical dislocation under anesthesia, and tumors were excised and weighed for further analysis.

### 2.6. Cell Lines and Cell Culture

The human cell lines Bel-7402, SK-Hep-1, Hep3B, PLC/PRF/5, and HEK-293T were acquired from the American Type Culture Collection (ATCC, Manassas, VA, USA). MHCC-97L, MHCC-97H, and HCC-LM3 were acquired from the Cell Resource Center, Institute of Basic Medical Sciences, CAMS/PUMC (Beijing, China). Cell lines were cultured in Dulbecco’s Modified Eagle Medium (DMEM) (Gibco, Carlsbad, CA, USA) or Minimum Essential Medium (MEM) medium (Gibco, Carlsbad, CA, USA), supplemented with 10% fetal bovine serum (FBS) (Gibco, Carlsbad, CA, USA) and 1% penicillin–streptomycin (Gibco, Carlsbad, CA, USA), at 37°C in a humidified 5% CO_2_ incubator.

### 2.7. Lentivirus Packaging Procedure and Infection

The plvx-HPX overexpression vector was obtained from Shanghai Generay Biotech Co., Ltd. (Shanghai, China). Lentivirus was produced in 293T cells using the pSPAX2/pVSV-G packaging system. Briefly, 293T cells were seeded in 10 cm dishes and transfected at 70–80% confluence with a plasmid mixture containing the transfer vector (plvx-HPX), pSPAX2, and pVSV-G (Addgene, Watertown, MA, USA) at a ratio of 4:3:1 using polyethylenimine (PEI) (Invitrogen, Waltham, MA, USA) in Opti-MEM (Gibco, Carlsbad, CA, USA). The medium was replaced with fresh complete DMEM 6–8 h post-transfection. Viral supernatants were harvested at 48 h post-transfection, centrifuged (3000 rpm, 10 min, 4 °C), filtered through 0.45 μm membranes, and stored at −80 °C. Target cells were infected at 30–50% confluence with viral supernatant containing 8 μg/mL Polybrene. After 12–16 h, the medium was replaced with fresh complete medium. Puromycin selection (1–2 μg/mL) was initiated 48 h post-infection and maintained for 3–7 days until complete death of control cells. Stable cell lines were validated by qPCR and/or Western blot analysis.

### 2.8. RNA Extraction and qRT-PCR

Total RNA was extracted from cells using TRIzol reagent (Invitrogen, #15596026CN, Waltham, MA, USA) according to the manufacturer’s protocol. RNA concentration and purity were determined by NanoDrop spectrophotometry (Thermo Fisher Scientific, Waltham, MA, USA). First-strand cDNA was synthesized from total RNA using Hifair^®^ III 1st Strand cDNA Synthesis SuperMix for qPCR (gDNA digester plus; Yeasen, #11141ES60, Shanghai, China), following the manufacturer’s instructions. Quantitative real-time PCR was performed using Hieff UNICON^®^ Universal Blue qPCR SYBR Green Master Mix (Yeasen, #11184ES08, Shanghai, China) on a Roche LightCycler^®^ 480 system. Each run included no-template controls for each primer pair. For each experimental condition, we performed ≥3 independent biological replicates, and each sample was measured in technical triplicate under the following conditions: 95 °C for 30 s, followed by 40 cycles of 95 °C for 10 s and 60 °C for 30 s. Gene expression levels were normalized to ACTB and calculated using the 2^−ΔΔCt^ method. The primers for real-time PCR were obtained from Tsingke (Beijing, China), and the sequences were as follows: *HPX* forward: CGTGACTGAACGCTGCTCA, *HPX* reverse: CTCCCGGTCCCATTTGTGAC; *ACTB* forward: GGACTTCGAGCAAGAGATGG, *ACTB* reverse: AGCACTGTGTTGGCGTACAG. The specificity of primers was confirmed by melt curve analysis.

### 2.9. Western Blotting

Cells were lysed in RIPA buffer (Beyotime, Shanghai, China) containing protease inhibitor cocktail on ice. Protein concentrations were quantified using the BCA assay (LABLEAD, Beijing, China). Equal amounts of protein (10–20 μg) were separated by SDS-PAGE (ACE Biotechnology, Changzhou, China) (140 V, 40 min) and transferred to 0.45 μm PVDF membranes (Merck Millipore, Darmstadt, Germany) using a wet transfer system (400 mA, 40–60 min). Membranes were blocked with 5% non-fat milk (Beyotime, Shanghai, China) for 1 h at room temperature and then incubated with primary antibodies overnight at 4 °C with the following primary antibodies: Caspase-3 (Rabbit monoclonal, 1:1000, ABclonal, #A19654, Wuhan, China), Bax (Rabbit monoclonal, 1:1000, ABclonal, #A19684), TNF-α (Rabbit polyclonal, 1:1000, ABclonal, #A11534, Wuhan, China), Hemopexin (HPX) (Rabbit monoclonal, 1:1000, ABclonal, #A9133, Wuhan, China), and Caspase-9 (Rabbit antibody, 1:1000, Abmart, #T55049, Shanghai, China). After washing with TBST (LABLEAD, Beijing, China), membranes were incubated with secondary antibody (HRP-conjugated Affinipure Goat Anti-Rabbit IgG (H + L), 1:5000, Proteintech, #SA00001-2, Wuhan, China) for 1h at room temperature. Membranes were then re-probed with GAPDH antibody (HRP-conjugated monoclonal, 1:5000, Proteintech, #HRP-60004, Wuhan, China) for 1 h. All protein bands were visualized using ECL reagent (ACE Biotechnology, Changzhou, China) and imaged with a chemiluminescence imaging system (Taono, Shanghai, China).

### 2.10. CCK-8 Assay

Cell proliferation was assessed using the CCK-8 assay (LABLEAD, #CK001, Beijing, China). Cells were digested, counted, and resuspended at a density of 5–8 × 10^4^ cells/mL. A 100 µL aliquot of the cell suspension was seeded into each well of a 96-well plate, with the peripheral wells filled with 100 µL PBS to minimize evaporation. Each experimental group included at least five replicate wells. After incubation, the medium was removed, and 100 µL of CCK-8 working solution (10% CCK-8 reagent in serum-free medium) was added to each well. Following 2 h incubation at 37 °C, the absorbance at 450 nm was measured using a microplate reader.

### 2.11. Clone Formation Assay

Cells were seeded into 6-well plates at a density of 500–1000 cells per well and cultured for 10–14 days to allow colony formation. Each experimental group included three replicate wells. After incubation, cells were washed with PBS, fixed with 4% paraformaldehyde (Solarbio, Beijing, China) for 15 min, and stained with 0.1% crystal violet for 20 min at room temperature. Colonies containing > 50 cells were counted manually under a microscope.

### 2.12. EdU Cell Proliferation Assay

Cell proliferation was evaluated using the BeyoClick™ EdU-594 Kit (Beyotime, #C0078S, Shanghai, China) according to the manufacturer’s protocol. Cells seeded in 24-well plates were incubated with 10 μM EdU for 2 h at 37 °C, fixed with 4% paraformaldehyde, and permeabilized with 0.3% Triton X-100. The Click reaction was performed using the fluorescent azide conjugate (594 nm excitation/615 nm emission) to label proliferating cells. Nuclei were counterstained with Hoechst (346 nm excitation/460 nm emission). Images were acquired using a fluorescence microscope (THUNDER Imaging Systems, Leica, Wetzlar, Germany), and EdU-positive cells were quantified using ImageJ software (v1.54g).

### 2.13. Wound Healing Assay

Cells were seeded in 6-well plates and cultured until reaching 90–100% confluence. A sterile 200 μL pipette tip was used to create a straight scratch wound in the monolayer. Detached cells were removed by washing with PBS, and fresh serum-free medium was added to minimize proliferation effects. The wounded area was photographed at 0 h and 24 h or 36 h under an inverted microscope.

### 2.14. Flow Cytometry Apoptosis Assay

Cell apoptosis was evaluated using the FITC Annexin V Apoptosis Detection Kit (BD Pharmingen™, #556547, San Diego, CA, USA) according to the manufacturer’s protocol. Cells were harvested, washed twice with cold PBS, and resuspended in 1×Annexin V binding buffer at a density of 1 × 10^6^ cells/mL. Cells were stained with FITC-conjugated Annexin V (5 µL/test) and 7-AAD (5 µL/test) for 15 min at room temperature in the dark. Apoptotic cells were analyzed by BD FACSAria™ II Workstation (BD, San Diego, CA, USA) within 1 h. Data were interpreted as follows: Annexin V−/7-AAD− (viable cells), Annexin V+/7-AAD− (early apoptotic cells), Annexin V+/7-AAD+ (late apoptotic cells). Statistical analysis was performed using FlowJo™ software (v10.8.1).

### 2.15. Mitochondrial Membrane Potential Assay Kit with JC-1 Analysis

Mitochondrial membrane potential was assessed using the JC-1 assay kit (Beyotime, #C2006, Shanghai, China) following the manufacturer’s protocol. Cells were seeded in 24-well plates and cultured under experimental conditions. After treatment, cells were incubated with JC-1 staining solution (2 μM) at 37 °C for 30 min in the dark. Following two washes with assay buffer, fluorescence was visualized using a THUNDER Imaging System (Leica, Wetzlar, Germany).

### 2.16. TUNEL Apoptosis Assay

The TUNEL assay was performed to detect apoptotic cells in paraffin-embedded tissue sections. Briefly, tissue sections were deparaffinized and rehydrated. After antigen retrieval in citrate buffer, sections were permeabilized with 0.1% Triton X-100. The TUNEL reaction mixture was applied and incubated at 37°C for 1 h. Nuclei were counterstained with DAPI (Beyotime, Shanghai, China). Fluorescence microscopy was performed with the following settings: TUNEL-positive cells (green): at 488/530 nm (excitation/emission), and DAPI (blue): at 358/461 nm (excitation/emission). Apoptotic cells were quantified by counting TUNEL-positive cells in at least 5 random fields per section and expressed as a percentage of total DAPI-stained nuclei.

### 2.17. H&E Staining and Immunohistochemistry Analysis

For histopathological evaluation, xenograft tumor tissue sections (4–5 μm) were stained with hematoxylin (Boster, Wuhan, China) and eosin (H&E) (Boster, Wuhan, China) following standard protocols as in a previous study [34]. For immunohistochemistry (IHC), sections were deparaffinized, rehydrated, and subjected to antigen retrieval. After blocking endogenous peroxidase activity and non-specific binding, sections were incubated overnight at 4°C with the following primary antibodies: HPX (Rabbit monoclonal, 1:100, ABclonal, #A9133), Bcl-2 (Rabbit monoclonal, 1:2000, ABclonal, #A19693), Caspase-3 (Rabbit monoclonal, 1:200, ABclonal, #A19654), and Bax (Rabbit monoclonal, 1:2000, ABclonal, #A19684). After washing, sections were incubated with HRP-conjugated goat anti-rabbit secondary antibody (1:200, Servicebio, #GB23303, Wuhan, China) for 1 h at room temperature. Staining was developed using DAB chromogen (Zhongshan Golden Bridge, #ZLI-9019, Beijing, China) and counterstained with hematoxylin. Two independent pathologists blinded to the experimental groups evaluated the stained sections using a semi-quantitative scoring system: Staining intensity (0: negative, 1: weak, 2: moderate, 3: strong) and percentage of positive cells (0: 0%, 1: 1–10%, 2: 11–50%, 3: 51–80%, 4: 81–100%). The final IHC score was calculated by multiplying the intensity score by the percentage score (range: 0–12). Representative images were captured by Pannoramic MIDI (3DHISTECH, Budapest, Hungary).

### 2.18. RNA-seq Analysis

RNA sequencing was performed by Tsingke (Beijing, China) using the Illumina NovaSeq X Plus Sequencing Systems. The raw sequencing data were aligned to the human reference genome GRCh38.112 using HISAT2 (v2.2.1), and gene expression levels were quantified as fragments per kilobase of transcript per million mapped reads (FPKM). Functional enrichment analyses were conducted for GO terms; Gene Set Enrichment Analysis (GSEA). The top significantly enriched terms/pathways (*p* < 0.05) were selected for further biological interpretation. All statistical analyses and visualizations were performed using Bioconductor packages.

### 2.19. Statistical Analysis

Statistical processing was performed using R v4.3.1 or GraphPad Prism v8.0.2. Continuous data from two groups were compared using unpaired two-tailed *t*-tests when distributions were approximately normal with comparable variances; otherwise, the Mann–Whitney U test was used. For three or more groups, we used one-way ANOVA under the same assumptions; when assumptions were not met, we used the Kruskal–Wallis test. Correlation analysis was conducted using the Pearson or Spearman methods. The Kaplan–Meier survival curve was used to distinguish the differences in survival time among patients in different groups. A *p* value < 0.05 was considered statistically significant.

## 3. Results

### 3.1. Analysis of the Biological Role of Fibrinolysis in Pan-Cancer

The fibrinolysis gene set was obtained from the GO database, with a total of 27 genes. This study first analyzed the correlations of these 27 genes at the transcriptome level in pan-cancer (Figure 1A), and the results suggested that there was a certain correlation in the expression among the genes. Subsequently, the CNV of 27 genes in pan-cancer samples were also analyzed, and it was found that *USF1* had a high amplification rate and a low deletion rate (Figure 1B). We evaluated the enrichment score of fibrinolysis at the pan-cancer level. The fibrinolysis enrichment score was relatively high in HCC (Figure 1C). The fibrinolysis enrichment score of liver tissue was higher than that of HCC, with a statistical difference. We also analyzed the effect of fibrinolysis on pan-cancer survival (Figure 1D). Compared with other tumors, fibrinolysis was a protective factor in HCC. Similarly, we analyzed the correlation between fibrinolysis and tumor stemness (Figure 1E) and found that fibrinolysis was negatively correlated with tumor stemness in HCC. The relationship between fibrinolysis and tumor death was also analyzed at the pan-cancer level (Figure 1F), indicating that fibrinolysis has a certain correlation with common patterns of tumor death. Based on the close relationship between fibrinolysis and pan-cancer biological characteristics mentioned above, we further analyzed the association between fibrinolysis and the mechanism of tumorigenesis and development at the pan-cancer level (Figure 1G). The heatmap shows that fibrinolysis has a strong positive correlation with COAGULATION, KRAS_SIGNALING, and ANGIOGENESIS.

### 3.2. Construction and Clinical Analysis of Fibrinolysis Molecular Subtypes in HCC

Through the analysis of fibrinolysis at the pan-cancer level, the fibrinolysis score in HCC was relatively high and was also a protective factor for the prognosis of HCC. For this reason, our subsequent research focused on HCC. Based on the 27 fibrinolysis genes and with a phenotypic correlation of 3, we classified the HCC samples into three subtypes (Figure 2A), and there were significant differences in prognosis among the three molecular subtypes (Figure 2B). Using a heatmap, we present the clinical characteristics of three fibrinolysis molecular subtypes and the differences in fibrinolysis gene expression (Figure 2C). At the level of tumorigenesis and development mechanisms, this study analyzed the differences of fibrinolysis molecular subtypes in the HALLMARK pathways (Figure 2D). In the tumor microenvironment, we found that, among the three fibrinolysis molecular subtypes, Cluster 2 had a relatively high ImmuneScore (Figure 2E), StromaScore (Figure 2F), and ESTIMATEScore (Figure 2G). In the analysis of the interaction between receptor and ligand pairs in the tumor microenvironment, we found that there were differences in the interactions of AFDN-EPHB6 (Figure 2H), EFNB1-EPHB6 (Figure 2I), and NTNG2-LRRC4 (Figure 2J). Based on the above analysis from clinical prognosis to the molecular mechanism level, it is suggested that fibrinolysis can classify HCC into three molecular subtypes with different characteristics.

### 3.3. A Risk Signature Was Constructed Based on the Fibrinolysis Molecular Subtype of HCC

Based on the significant characteristics of the fibrinolysis molecular subtypes, we further constructed a prognostic signature. Firstly, we analyzed the differentially expressed genes (DEGs) between HCC and normal liver tissues in TCGA-LIHC. Based on the thresholds of adjusted *p* value < 0.05 and the absolute value of log_2_foldchange > 1, a total of 4548 DEGs were obtained. We included different subtypes in WGCNA and correlated them with the transcriptome data of DEGs, obtaining a total of 15 gene modules (Figure 3A). Among them, the blue module had the greatest correlation with molecular typing (Figure 3B). By conducting Kyoto encyclopedia of genes and genomes (KEGG) analysis on the blue module (Figure 3C), it was found that the blue module was involved in important biological functions, such as the PPAR signaling pathway, drug metabolism-cytochrome P450, and chemical carcinogenesis-DNA adducts. Based on the preset threshold, we extracted 62 genes from the blue module for constructing the risk signature (Figure 3D).

We randomly split the TCGA-LIHC cohort (follow-up > 0) into training and internal test sets (5:5). Fibrinolysis-related signatures were first screened by univariable Cox analysis, and then LASSO-Cox was performed in the training set (Figure 3E,F). After multivariable Cox regression, six genes (*ACAT1*, *GRHPR*, *HPX*, *PCK2*, *IYD*, and *PON1*) were retained to construct the prognostic signature. For each patient, a risk signature was calculated as follows:(2)$$\begin{array}{c} RiskScore=h_{0}(t)exp(-0.3320497\times ACAT1-0.5356651\times GRHPR-\\ 0.2373711\times HPX+0.5909421\times PCK2-0.2974011\times IYD-\\ 0.1330290\times PON1). \end{array}$$

Grouping was based on the median RiskScore of each dataset. This study verified the risk signature in the training set and the entire set. In the training set, survival analysis indicated that the survival outcome in low-risk group patients was significantly better (Figure 3G). The result of the ROC curve (Figure 3H) suggested that the risk signature had a predictive ability (1-year AUC = 0.801; 3-year AUC = 0.781; 5-year AUC = 0.820). In the entire set, this signature also had a good predictive ability (Figure 3I,J). The research was verified by using LIRI-JP (Figure 3K) and the GSE54236 dataset (Figure 3L) from different sources as external validation sets. It can also be observed that patients in the low-risk group have a survival advantage. Univariate (Figure 3M) and multivariate (Figure 3N) Cox regression analyses were conducted in combination with relevant clinical variables, both of which indicated that the RiskScore was significantly correlated with the survival of patients (*p* < 0.05).

### 3.4. The Key Gene HPX in HCC Was Identified Through the Risk Signature

Through the risk signature, we obtained six genes related to fibrinolysis molecular typing. At the level of DEG, we found that the absolute value of log_2_foldchange of *HPX* was the largest (Figure 4A). At the tumor microenvironment level, we found that *HPX* was closely related to hepatocytes, adipocytes, and endothelial cells (Figure 4B) and may play an important biological role in HCC. At the level of correlation with fibrinolysis, the result suggested that the correlation coefficient of *HPX* was the largest (Figure 4C). Meanwhile, it is a DEG in HCC tissues, and its expression is relatively high in liver tissues at the pan-cancer level (Figure 4D). At the prognostic level, patients with high *HPX* expression had a relatively better prognosis (Figure 4E). Therefore, our research found that *HPX* may be a key gene in HCC, and subsequent experimental studies focused on *HPX*.

### 3.5. HPX Overexpression Suppressed Proliferation and Migration in HCC Cells

To investigate the role of *HPX* in HCC, we established *HPX*-overexpressing cell lines. Baseline *HPX* mRNA levels varied across HCC cell lines, with Bel-7402, HCC-LM3, and MHCC-97L exhibiting relatively low expression (Figure 5A). Successful overexpression of *HPX* was confirmed by RT-qPCR in *HPX*-overexpressed cells compared to control cells in the Bel-7402 (Figure 5B), HCC-LM3 (Figure 5C), and MHCC-97L (Figure 5D) lines. Functional assays demonstrated that *HPX* overexpression substantially inhibited cell proliferation, as evidenced by reduced colony formation in Bel-7402 (Figure 5E), HCC-LM3 (Figure 5F), and MHCC-97L (Figure 5G) cells along with decreased EdU incorporation in Bel-7402 (Figure 5H), HCC-LM3 (Figure 5I), and MHCC-97L (Figure 5J) cells. Additionally, wound healing assays revealed that *HPX*-overexpressed cells showed delayed migration compared to controls in Bel-7402 (Figure 5K), HCC-LM3 (Figure 5L) and MHCC-97L (Figure 5M) cells, indicating suppression of proliferation. Consistently, the CCK-8 assay demonstrated significantly reduced cell viability in the *HPX*-overexpressing groups across all three cell lines (Figure 5N). Collectively, these data indicate that *HPX* functions as a suppressor of proliferation and migration in HCC cells.

### 3.6. HPX Overexpression Promoted Apoptosis Through Mitochondrial Pathway Activation

To determine whether *HPX* overexpression influences apoptosis in HCC cells, we analyzed apoptosis rates by flow cytometry. Compared to NC cells, *HPX*-overexpressed cells exhibited significantly higher proportions of apoptotic cells in the Bel-7402, HCC-LM3, and MHCC-97L lines (Figure 6A), supporting a pro-apoptotic role. Mitochondrial membrane potential assessment using JC-1 staining showed increased depolarization in HPX-overexpressing cells, evidenced by decreased red aggregates and increased green monomers across all three cell lines (Figure 6B), further supporting mitochondrial involvement in *HPX*-induced apoptosis. In order to further explore the mechanism of how *HPX* promoted apoptosis, we performed RNA-seq in *HPX*-overexpressed MHCC-97L cells. Transcriptomic analysis and GO enrichment suggested activation of heme transport, hemoglobin metabolic process, and heme transmembrane transporter activity pathways following *HPX* overexpression (Figure 6C), consistent with the biological role of *HPX*. GSEA results showed *HPX* might promote apoptosis via TNF-α signaling pathways (Figure 6D). Next, Western blotting confirmed molecular changes characteristic of apoptosis: increased expression of Caspase-9 and Caspase-3 and upregulation of pro-apoptotic Bax and TNF-α in *HPX*-overexpressing cells compared to controls across all cell lines (Figure 6E). These findings demonstrate that *HPX* induces apoptosis in HCC cells by activating the mitochondrial apoptotic pathway.

### 3.7. HPX Overexpression Inhibited Tumor Growth and Promoted Apoptosis In Vivo

To evaluate the impact of *HPX* overexpression on tumor growth in vivo, xenograft models were established using *HPX*-overexpressed and NC cells. Tumors derived from *HPX*-overexpressed cells were significantly smaller in size compared to controls, as shown by representative images, indicating that *HPX* inhibits tumor growth in vivo (Figure 7A). Tumor volume measurements over time revealed that *HPX* overexpression markedly suppressed tumor growth (Figure 7B,E), which was further supported by the significantly reduced tumor weights observed at endpoint (Figure 7C). Consistent with these findings, TUNEL staining demonstrated increased apoptotic activity in tumors from the *HPX*-overexpressed group relative to NC (Figure 7D). Histological analysis indicated typical tumor morphology (H&E staining) alongside enhanced expression of the pro-apoptotic marker Caspase-3 and Bax and decreased expression of the anti-apoptotic protein Bcl-2 in *HPX*-overexpressed tumors compared to controls (Figure 7F upper). Quantification of immunoreactivity scores corroborated these alterations (Figure 7F below). Collectively, these results demonstrate that *HPX* overexpression suppresses tumor growth and promotes apoptosis through modulation of apoptosis-related proteins in vivo.

## 4. Discussion

This study systematically investigated the role of fibrinolysis in pan-cancer; we successfully identified the fibrinolysis-related gene *HPX* influencing HCC progression. Functional experiments demonstrated that *HPX* aligned with suppression of tumor phenotypes. Clinically, as a secreted protein, *HPX* has biomarker potential for risk assessment. These findings nominate *HPX* as a candidate for clinical stratification and therapeutic exploration while underscoring the need for validation in larger, etiologically diverse cohorts and for pathway-level perturbation studies to define causality.

The bioinformatics methods employed in this study, including differential expression analysis, molecular subtyping, and the risk signature construction, proved effective in uncovering the clinical relevance of fibrinolysis in HCC. Biologically, fibrinolysis can modulate tumor progression through extracellular matrix remodeling, angiogenesis, and metastatic dynamics; clinical practice also underscores this axis, as low-molecular-weight heparin (LMWH) is widely used for cancer-associated thrombosis and may influence the coagulation–fibrinolysis balance [35]. Preclinical microsatellite stable colorectal cancer models demonstrate that LMWH can synergize with adoptive cell transfer or anti-PD-1, improving tumor vascular normalization and CD8⁺ T-cell infiltration to suppress growth and liver metastasis [36]. While many previous studies have focused on well-known fibrinolytic factors such as uPA, uPAR, and PAI-1, our study highlights *HPX* as a novel and clinically significant player in HCC. *HPX*, traditionally known for its role in heme scavenging, has emerging evidence linking it to cancer progression [37,38,39]. Our risk signature confirmed its prognostic value, and experimental validation further supported its tumor-suppressive function via apoptosis induction. Given its association with patient survival and tumor behavior, *HPX* is associated with protective outcomes and a potential therapeutic target in HCC.

Published research has revealed that *HPX* has been linked to disease progression in pancreatic ductal adenocarcinoma (PDAC)—stromal *HPX* expression correlated with lymph node metastasis, suggesting that *HPX* may serve as a biomarker or potential therapeutic target in PDAC [40]—while fucosylated hemopexin showed significantly higher values in samples from patients with cirrhosis and HCC than hepatitis or normal individuals [41,42]. However, the functions of *HPX* are complex and exhibit tumor-type-specific heterogeneity. In prostate cancer, *HPX* levels are decreased in both patient plasma and tumor stroma, with low expression correlating with poor prognosis and increased tumor volume [43]. In this study, through pan-cancer analysis, we found that *HPX* is lowly expressed in HCC and negatively correlated with poor prognosis. Furthermore, we elaborated on the potential of the expression of *HPX* in HCC as a biomarker.

The relationship between fibrinolysis and cell death, particularly apoptosis, remains an underexplored area in cancer biology. Our study provides compelling evidence that fibrinolysis-related genes, especially *HPX*, play a crucial role in regulating apoptotic pathways in HCC. Both in vitro and in vivo experiments demonstrated that *HPX* overexpression significantly enhances apoptosis, thereby suppressing tumor growth and metastatic potential. These findings align with recent studies suggesting that fibrinolytic components can modulate cell survival and death signals, but our work specifically establishes *HPX* as a key mediator of apoptosis in HCC. In contrast, previous studies on *HPX* have mainly focused on its role as a scavenger of labile heme in the tumor microenvironment, where it suppresses cancer progression by limiting heme-induced oncogene activation, such as c-MYC [43]. While those studies emphasize *HPX*’s function in regulating tumor proliferation and metastasis through heme clearance, our research highlights its direct involvement in promoting apoptosis in HCC, indicating potentially distinct but complementary mechanisms. Further mechanistic studies are warranted to elucidate the precise signaling pathways through which *HPX* exerts its pro-apoptotic effects.

Many Asian patients with HCC have underlying chronic hepatitis B or C infection and cirrhosis and experience ‘rebalanced’ hemostasis due to impaired hepatic synthesis, portal hypertension–related thrombocytopenia, and inflammation. These changes can shift coagulation–fibrinolysis dynamics and confound circulating markers, including HPX. Accordingly, clinical translation of HPX and fibrinolysis-based signatures should account for liver function, etiology (HBV/HCV, alcohol-related, NASH), and anticoagulant exposure. Future validation will stratify analyses by these factors and employ standardized preanalytical handling to ensure interpretability across diverse patient populations. To strengthen clinical applicability, multi-center validation with prespecified endpoints in larger and etiologically diverse cohorts is needed.

Despite these significant findings, our study has several limitations. First, while we identified *HPX* as a critical regulator of apoptosis in HCC, strong relationship data between HPX and TNF-α-Bax/Bcl-2 are lacking. Second, although our risk signature demonstrated strong prognostic value, external validation in independent cohorts is needed to confirm its clinical applicability. Third, our in vivo evidence comes from xenografts in immunodeficient mice that lack cirrhosis; responses may differ in immunocompetent or fibrotic livers. Etiologic heterogeneity in HCC (HBV/HCV, alcohol-related, NASH) and rebalanced hemostasis can modify fibrinolysis programs and HPX biology. Because HPX is secreted, circulating levels may be influenced by liver function, hemolysis, inflammation, and preanalytical handling.

## 5. Conclusions

In summary, our study provides a comprehensive analysis of fibrinolysis in pan-cancer and establishes its critical role in HCC progression. We identified three novel molecular subtypes of HCC based on fibrinolysis and developed a prognostic risk signature with potential clinical utility. Importantly, *HPX* was associated with a tumor-suppressive phenotype and increased apoptosis, consistent with TNF-α-mediated mitochondrial signaling, offering new insights into HCC pathogenesis and therapy. These findings pave the way for further exploration of fibrinolysis-related mechanisms in cancer and highlight *HPX* as a promising target for precision medicine in HCC.

## Figures and Tables

**Figure 1 cancers-17-02969-f001:**
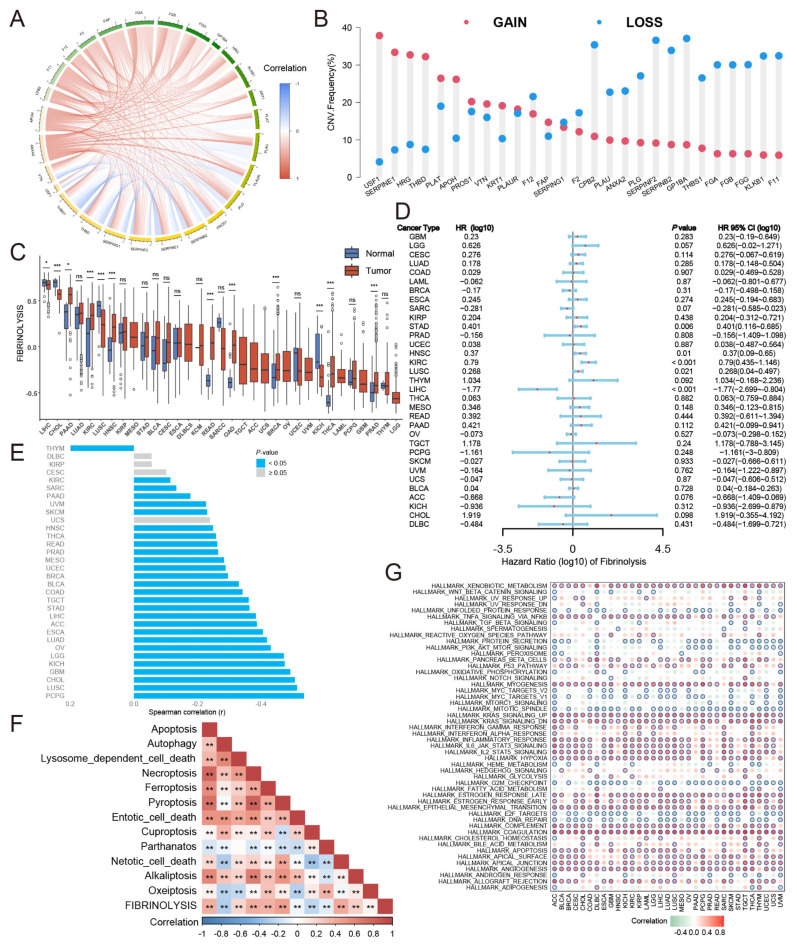
Pan-cancer analysis of fibrinolysis. (**A**) Correlation analysis of fibrinolysis-related genes across cancer types. (**B**) Copy number variation (CNV) landscape of fibrinolysis genes in pan-cancer datasets. (**C**) Pan-cancer analysis of fibrinolysis scores. (**D**) Univariate Cox regression analysis evaluating the prognostic significance of fibrinolysis. (**E**) Association between fibrinolysis expression and tumor stemness indices. (**F**) Correlation of fibrinolysis with distinct tumor cell-death programs. (**G**) Enrichment analysis linking fibrinolysis to HALLMARK oncogenic pathways (the blue circle indicates that the absolute value of the correlation coefficient is greater than 0.3 and the *p* value is less than 0.05). * *p* < 0.05, ** *p* < 0.01, *** *p* < 0.001, ns: no significance.

**Figure 2 cancers-17-02969-f002:**
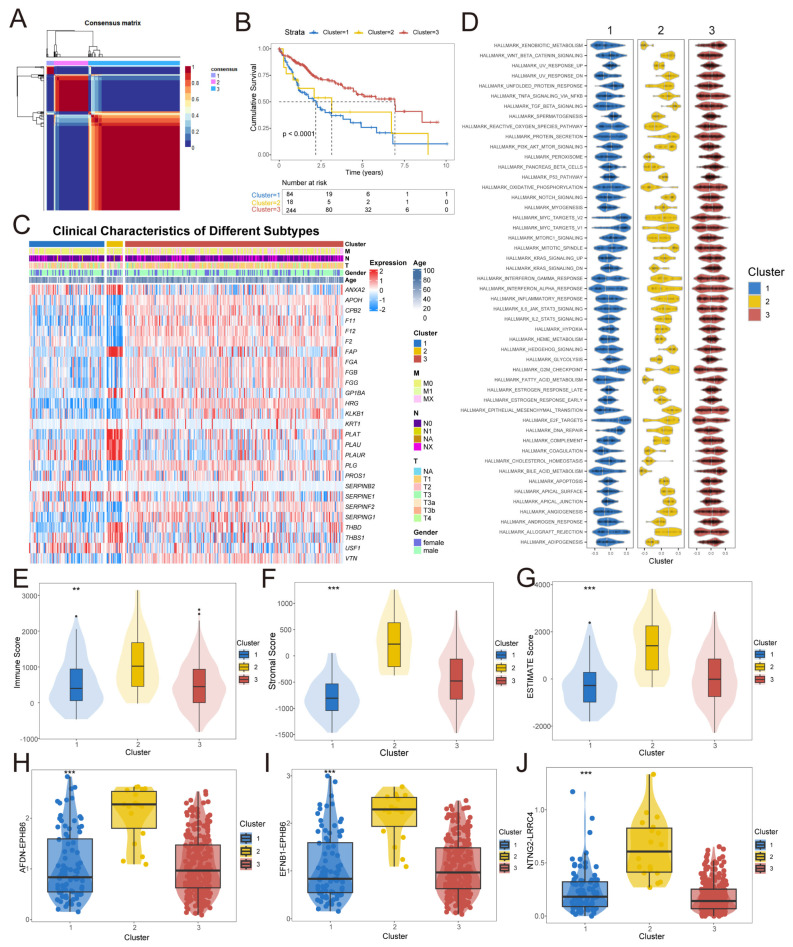
Molecular subtyping and risk signature construction of HCC based on fibrinolysis. (**A**) Identification of three distinct HCC molecular subtypes via non-negative matrix factorization (NMF). (**B**) Survival analysis of HCC molecular subtypes stratified by fibrinolysis. (**C**) Heatmap displaying clinical characteristics across HCC molecular subtypes. (**D**) HALLMARK pathway enrichment analysis of HCC subtypes. (**E**–**G**) Tumor microenvironment (TME) assessment of subtypes using ImmuneScore, StromalScore, and ESTIMATEScore. (**H**–**J**) Receptor–ligand interaction analysis in the TME (AFDN-EPHB6, EFNB1-EPHB6, and NTNG2-LRRC4). ** *p* < 0.01, *** *p* < 0.001.

**Figure 3 cancers-17-02969-f003:**
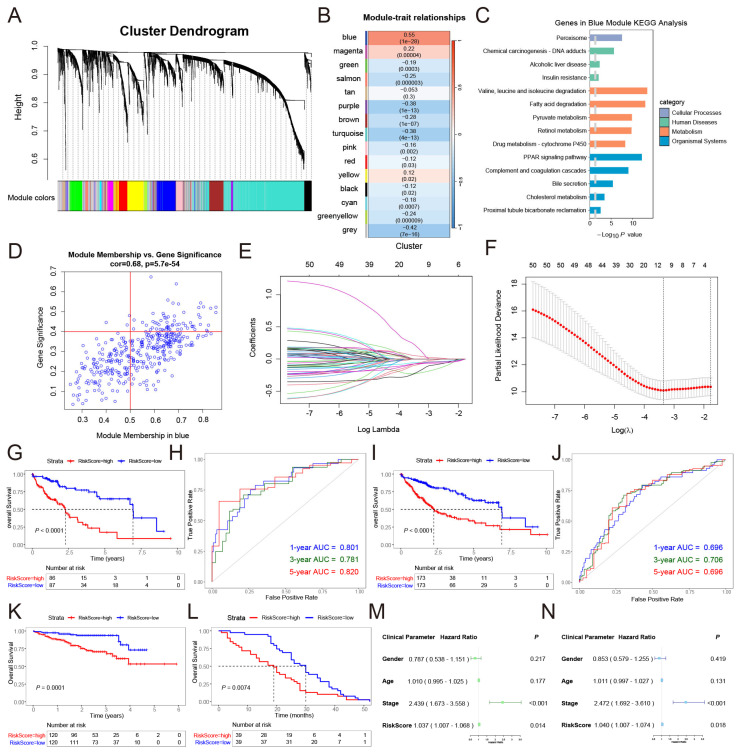
Construction of a risk signature based on fibrinolysis molecular subtypes. (**A**) Identification of gene modules by weighted gene co-expression network analysis (WGCNA), and different colors represent different modules. (**B**) Correlation analysis between gene modules and molecular subtypes. (**C**) KEGG pathway enrichment analysis of the blue module, and the grey dashed line represents −log_10_*P*(0.05). (**D**) Screening of hub genes in the blue module, and the blue circles represent different genes. (**E**,**F**) Further gene selection using LASSO regression. The dotted line on the left represents lambda.min, and the dotted line on the right represents lambda.1se. (**G**) Survival analysis of the training cohort. (**H**) Time-dependent receiver operating characteristic (ROC) curve analysis (AUC) of the training cohort. (**I**) Survival analysis of the entire cohort. (**J**) AUC evaluation of the entire cohort. (**K**,**L**) External validation of the risk signature using the LIRI-JP and GSE54236 datasets. (**M**,**N**) Univariate and multivariate Cox regression analyses of the risk signature in TCGA-LIHC.

**Figure 4 cancers-17-02969-f004:**
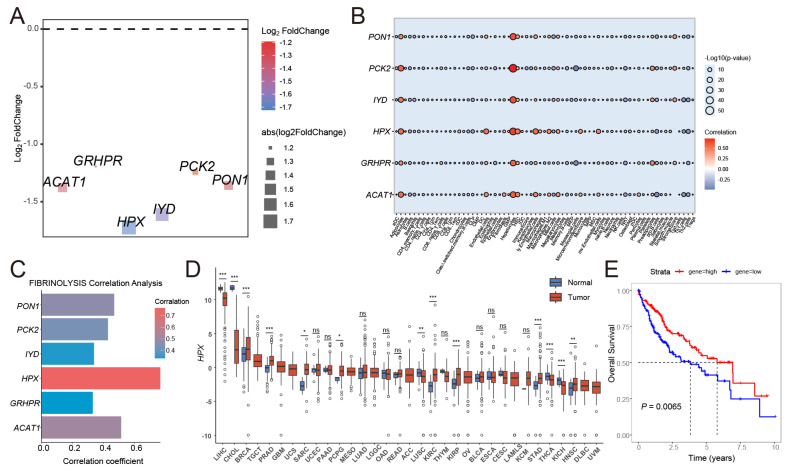
Identification of *HPX* as a key gene in HCC. (**A**) Scatter plot showing the distribution of six candidate genes. (**B**) Correlation analysis between the six candidate genes and tumor microenvironment-infiltrating immune cells in HCC. (**C**) Association of the six candidate genes with fibrinolysis scores in HCC. (**D**) Pan-cancer differential expression analysis of *HPX*. (**E**) Prognostic impact of HPX expression in HCC patients. * *p* < 0.05, ** *p* < 0.01, *** *p* < 0.001, ns: no significance.

**Figure 5 cancers-17-02969-f005:**
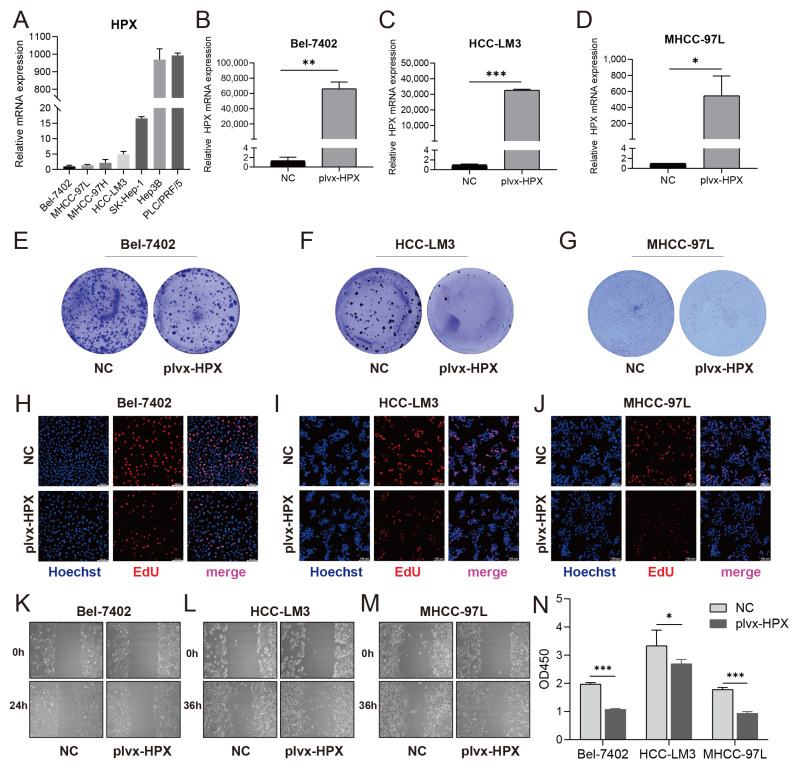
*HPX* overexpression suppressed proliferation and migration in HCC cells. (**A**) Relative mRNA expression levels of *HPX* in multiple cell lines. (**B**–**D**) *HPX* mRNA expression quantified by qRT-PCR in Bel-7402, HCC-LM3, and MHCC-97L cells transfected with plvx-HPX or negative control (NC). (**E**–**G**) Colony formation assays of Bel-7402, HCC-LM3, and MHCC-97L cells. (**H**–**J**) EdU staining in Bel-7402, HCC-LM3, and MHCC-97L cells. (**K**–**M**) Wound healing assays over indicated times (0 h, 24 h, or 36 h) in Bel-7402, HCC-LM3, and MHCC-97L cells. (**N**) Quantitative analysis of cell viability by OD450 measurement across the three cell lines. * *p* < 0.05, ** *p* < 0.01, *** *p* < 0.001.

**Figure 6 cancers-17-02969-f006:**
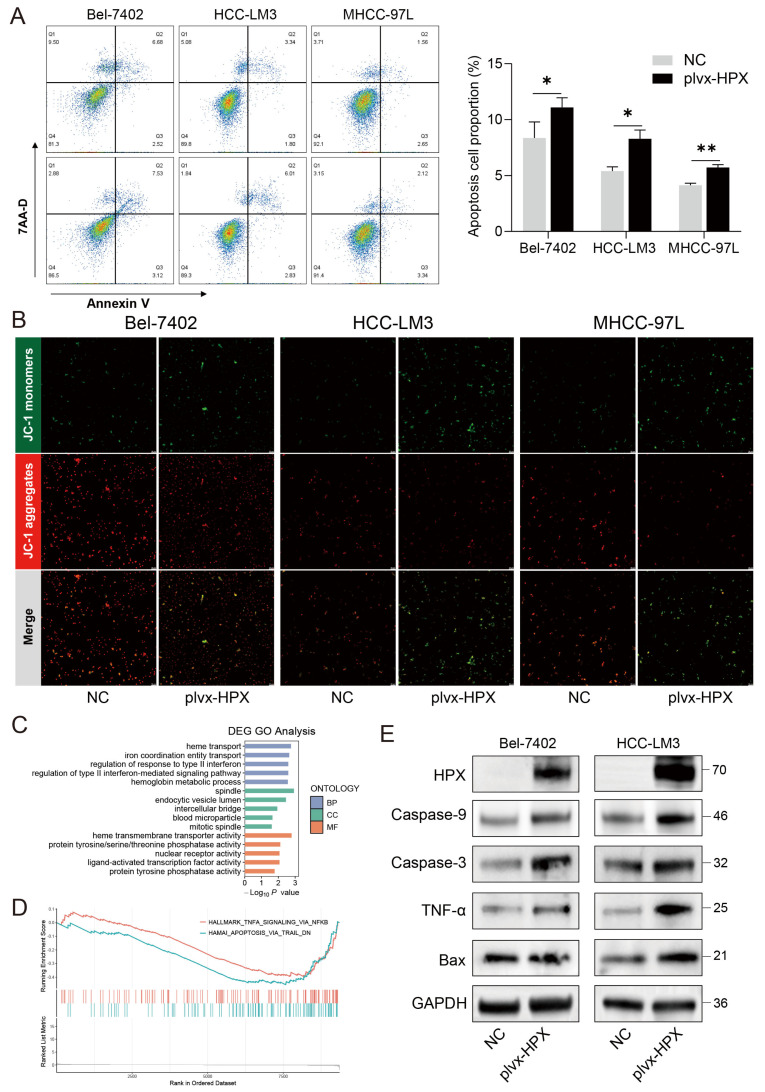
HPX promoted apoptosis through mitochondrial pathway in vitro. (**A**) Flow cytometry analysis of apoptosis in Bel-7402, HCC-LM3, and MHCC-97L cells transfected with plvx-HPX or NC. (**B**) JC-1 staining indicating mitochondrial membrane potential in Bel-7402, HCC-LM3, and MHCC-97L cells. (**C**) GO enrichment analysis of DEGs categorized into BP, CC, and MF. (**D**) GSEA analysis in the *HPX*-overexpressed HCC cells. (**E**) Western blot analysis of protein expression levels for apoptosis pathways (Appendix A). * *p* < 0.05, ** *p* < 0.01.

**Figure 7 cancers-17-02969-f007:**
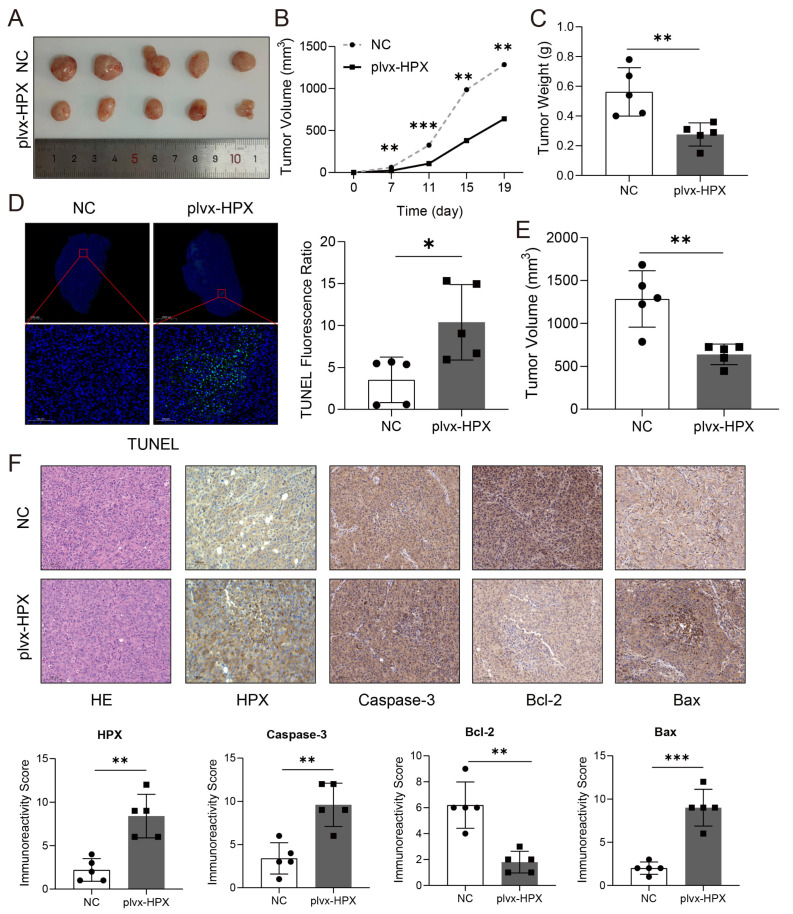
*HPX* inhibited tumor growth and promotes apoptosis in vivo. (**A**) Tumors derived from MHCC-97L cells transfected with plvx-HPX (*n* = 5) or NC (*n* = 5). (**B**) Tumor growth curves recorded over 19 days post-implantation. (**C**) Tumor weights measured at endpoint from the plvx-HPX and NC groups. (**D**) TUNEL staining images and quantification of apoptosis in plvx-HPX and NC groups. (**E**) Tumor volume measurements in an additional experimental cohort comparing NC and plvx-HPX groups. (**F**) Histological analysis of tumor sections stained with HE and immunohistochemical detection of HPX, Caspase-3, Bcl-2, and Bax proteins, alongside corresponding immunoreactivity scores (* *p* < 0.05, ** *p* < 0.01, *** *p* < 0.001).

## Data Availability

The data analyzed in the current study are available from the corresponding author upon reasonable request.

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
