# Peer review of "Hemopexin Suppresses Hepatocellular Carcinoma via TNF-α-Mediated Mitochondrial Apoptosis"

_cancers, 2025, doi:10.3390/cancers17182969_

Round 1
Reviewer 1 Report
Comments and Suggestions for Authors
The manuscript addresses an important topic within the field of medicinal chemistry and drug design. The topic is relevant, and the experimental approach is of potential interest to the readership of Cancers. However, several issues regarding methodological rigor, data interpretation, and presentation must be addressed before the manuscript can be considered for publication.
Title: In my opinion, the title is too long and complicated. It should be shortened to make it clearer. Abstract section: (i) The abstract is too long and not prepared in accordance with mdpi requirements. It should not exceed 200 words and should be one paragraph, excluding the words "Background," etc., and should also include this information. (ii) Some fragments in the abstract sound unnatural, e.g., "Fibrinolysis is highly expressed in liver tissues and serves as a protective factor..." (page 2); it should be "...and serves as a protective factor…"; "could upregulated Bax/Bcl-2 expression" (p. 2) is a grammatical error – correctly: "could upregulate Bax/Bcl-2 expression." (iii) The abstract uses too many abbreviations – a reader unfamiliar with the context may have difficulty understanding this abstract. Please correct this. Introduction section: (i) In some places, the style is chaotic and not very fluid: "Despite continuous advancements in cancer diagnosis and treatment strategies are constantly being updated..." (ii) References to the literature are correct, but some information requires clarification (e.g., the lack of a full description of the mechanisms for fibrinogen/FpA (pages 2–3)). (iii) There is no clear presentation of the research hypothesis. Please clearly state what the research hypothesis was, what the authors wanted to achieve, test, etc. Methods section: (i) The description is detailed, but in places too technical, without justification. Examples for improvement: "Mice were euthanized humanely by the method cervical dislocation under anesthesia" (p. 4) – awkward wording; it should be: "...by cervical dislocation under anesthesia." (ii) In the qPCR methods (p. 4) – there is no information about the negative control and biological replicates (only technical triplicates). (iii) The statistical analysis includes only general description (t-test, ANOVA), but no justification for the selection of tests under specific conditions. (iv) Explain the meaning of the individual symbols in the equation on page 3, line 127. Results section: (i) Figures have very small fonts, making them difficult or even impossible to read. (ii) The legends for some in vivo experiments lack precise descriptions of sample sizes (n). (iii) The result descriptions repeat what is seen in the graphs rather than interpreting them. For example: "Tumor volume was significantly smaller in the HPX-overexpressed group (Fig. 7B)" – it would be appropriate to immediately add "indicating that HPX inhibits tumor growth in vivo." (iv) In the bioinformatics analysis (pp. 11–12): the description of the risk model and regression coefficients is too technical – it would be worthwhile to simplify it. Discussion section: It is worth highlighting a strength of the discussion: the context of fibrinolysis and the role of HPX is well presented. Notes: (i) In this section, in many places, the authors repeated the results from the Results section rather than interpreting them. (ii) There is no clear comparison with other studies that indicate HPX as a progression promoter in some cancers (e.g., PDAC – mentioned, but not elaborated upon). (iii) Furthermore, I believe that this section lacks a clear emphasis on potential translational limitations, e.g., differences between mouse models and patients. These limitations are mentioned (p. 20), but are very general. Substantive errors: (i) The abstract states that fibrinolysis "serves as a protective factor for HCC across pan-cancers." This wording is imprecise. It should be: "...is associated with protective outcomes in HCC compared to other cancers." (ii) In my opinion, the HPX–TNF-α–Bax/Bcl-2 relationship is inferred primarily based on correlation and expression; however, strong causal data are lacking (e.g., no experiments with TNF-α inhibitors). (iii) In the in vivo analyses (pp. 18–19), there is no clear information on the number of animals in each group; it is difficult to assess statistical reliability. After considering all my comments and suggestions, and after making corrections and additions, the manuscript may be reconsidered for submission, provided that the other reviewers also agree. Status: major revision.
Comments on the Quality of English Language
Style and language: the manuscript requires very careful editing by a native speaker well-versed in the technical vocabulary used in the manuscript. (i) The text contains numerous grammatical and stylistic errors, e.g., "Further research is needed essential to elucidate..." (should be: "Further research is essential to elucidate..."), "HPX could upregulated of Bax/Bcl-2..." (should be "HPX could upregulate Bax/Bcl-2..."), method cervical dislocation under anesthesia", etc. (ii) Please pay close attention to the tenses, e.g., some methods are described in the past tense, others in the present tense. (ii) Please explain all abbreviations where they are used for the first time (e.g., "FpA" appears without prior explanation in the main text, only in the abstract).
Author Response
|
Comments 1: Title: In my opinion, the title is too long and complicated. It should be shortened to make it clearer. |
|
Response 1: We thank the reviewer for this helpful suggestion and fully agree that the title should be shorter and clearer. Revised title: Hemopexin suppresses hepatocellular carcinoma via TNF-α–mediated mitochondrial apoptosis.
|
|
Comments 2: Abstract section: (i) The abstract is too long and not prepared in accordance with mdpi requirements. It should not exceed 200 words and should be one paragraph, excluding the words "Background," etc., and should also include this information. |
|
Response 2: We sincerely thank the reviewer for this precise guidance and fully agree to comply with MDPI’s requirements. We have rewritten the Abstract as a single paragraph not exceeding 200 words: Abstract: Fibrinolysis maintains coagulation homeostasis, yet its roles in hepatocellular carcinoma (HCC) are incompletely defined. We developed a fibrinolysis-based molecular classification and prognostic signature for HCC and to identify a key regulatory gene. Using non-negative matrix factorization (NMF), we defined fibrinolysis-related HCC subtypes with distinct outcomes and tumor-microenvironment features. Weighted gene co-expression network analysis (WGCNA) plus least absolute shrinkage and selection operator (LASSO) regression identified a six-gene signature (ACAT1, GRHPR, HPX, PCK2, IYD, PON1) that stratified risk across cohorts. Hemopexin (HPX) emerged as the lead candidate and was functionally validated. HPX overexpression reduced clonogenicity and migration, increased apoptosis, and restrained xenograft growth. RNA-seq suggest that HPX is associated with apoptosis and TNF-α signaling pathways, flow cytometry apoptosis assay, mitochondrial membrane potential assay and terminal deoxynucleotidyl transferase dUTP nick end labeling (TUNEL) staining and were performed to validate these findings above. Western blot and immunohistochemistry showed HPX could upregulate the Bax/Bcl-2 ratio via TNF-α signaling pathway. This study identified novel molecular subtypes of HCC. HPX was associated with an anti-tumor phenotype, consistent with TNF-α–mediated mitochondrial apoptosis, including an increased Bax/Bcl-2 ratio. (Page 1-2. Line 38-54)
Comments 3: (ii) Some fragments in the abstract sound unnatural, e.g., "Fibrinolysis is highly expressed in liver tissues and serves as a protective factor..." (page 2); it should be "...and serves as a protective factor…"; "could upregulated Bax/Bcl-2 expression" (p. 2) is a grammatical error – correctly: "could upregulate Bax/Bcl-2 expression." Response 3: We thank the reviewer for these precise language corrections and fully agree. We have revised the offending phrases and systematically edited similar expressions across the manuscript for grammar, style, and scientific precision. In revised manuscripts, we have removed the sentences: Fibrinolysis is highly expressed in liver tissues and serves as a protective factor...; and the grammatical errors in the abstract have been corrected: “could upregulate the Bax/Bcl-2 ratio via TNF-α signaling pathway. (Page 2. Line 51-52)
Comments 4: (iii) The abstract uses too many abbreviations – a reader unfamiliar with the context may have difficulty understanding this abstract. Please correct this. Response 4: We thank the reviewer for this clear guidance and fully agree. We have minimized abbreviations to only those essential for comprehension and defined them at first mention. The revised abstract could be found in response to comments 2.
Comments 5: Introduction section: (i) In some places, the style is chaotic and not very fluid: "Despite continuous advancements in cancer diagnosis and treatment strategies are constantly being updated..." Response 5: We thank the reviewer for this helpful observation and fully agree. We have polished the opening for clarity and flow. The revised sentence is: 1. Despite continuous advances in cancer diagnosis and treatment, outcomes for hepatocellular carcinoma (HCC) remain unsatisfactory, underscoring the need for mechanistically grounded biomarkers and therapeutic targets.” (Page 2, line 65-67)
Comments 6: (ii) References to the literature are correct, but some information requires clarification (e.g., the lack of a full description of the mechanisms for fibrinogen/FpA (pages 2–3)). Response 6: We thank the reviewer for this helpful suggestion and fully agree to add a concise mechanistic note. We have simplified the description as follows: Basically, fibrinogen is cleaved by thrombin to release fibrinopeptides A and B (FpA/FpB), exposing polymerization sites that drive fibrin assembly; factor XIIIa then cross-links fibrin and stabilizes the clot. Fibrinolysis counterbalances this process: tissue-type or urokinase-type plasminogen activator converts plasminogen to plasmin, which degrades cross-linked fibrin into D-dimer and related fragments [4]. (Page 2, line 69-74) Reference 4. Weisel JW, Litvinov RI. Mechanisms of fibrin polymerization and clinical implications. Blood.2013; 121(10):1712-1719.https://doi.org/10.1182/blood-2012-09-306639.
Comments 7: (iii) There is no clear presentation of the research hypothesis. Please clearly state what the research hypothesis was, what the authors wanted to achieve, test, etc. Response 7: We thank the reviewer for this important suggestion and fully agree that the research hypothesis should be stated explicitly. We have replaced the following paragraph to the last paragraph of Introduction to make our hypothesis and objectives clear: 1. On this basis, this study aims to investigate the role of fibrinolysis in HCC. We then develop and validate a prognostic signature derived from fibrinolysis-related characteristics and examine its clinical relevance and prognostic value. Finally, we nominate a priority candidate gene for functional evaluation and explore pathways that may underlie its effects. (Page 2-3, line 93-97)
Comments 8: Methods section: (i) The description is detailed, but in places too technical, without justification. Examples for improvement: "Mice were euthanized humanely by the method cervical dislocation under anesthesia" (p. 4) – awkward wording; it should be: "...by cervical dislocation under anesthesia." Response 8: We thank the reviewer for this helpful suggestion and fully agree to simplify the wording and provide brief justifications where appropriate. We have streamlined the Methods and corrected the euthanasia sentence as follows: 1. mice were euthanized humanely by cervical dislocation under anesthesia. (Page 4, line 150-151)
Comments 9: (ii) In the qPCR methods (p. 4) – there is no information about the negative control and biological replicates (only technical triplicates). Response 9: We thank the reviewer for this important point and fully agree. We have added the following details to ensure reproducibility and clarity: 1. Each run included no-template controls for each primer pair. For each experimental condition, we performed ≥ 3 independent biological replicates, and each sample was measured in technical triplicate under the following conditions. (Page 4, line 180-183)
Comments 10: (iii) The statistical analysis includes only general description (t-test, ANOVA), but no justification for the selection of tests under specific conditions. Response 10: We thank the reviewer for this helpful suggestion and agree to state the usage scenarios concisely. We have clarified the decision rules for the tests actually used as follows: 1. Continuous data from two groups were compared using unpaired two-tailed t-tests when distributions were approximately normal with comparable variances, otherwise, the Mann–Whitney U test was used. For three or more groups. For three or more groups, we used one-way ANOVA under the same assumptions; when assumptions were not met, we used the Kruskal–Wallis test. (Page 7, line 290-295)
Comments 11: (iv) Explain the meaning of the individual symbols in the equation on page 3, line 127. Response 11: We thank the reviewer for this clear request and fully agree. The meaning of every symbol was as follows: 1. : baseline hazard (baseline risk) at time t; 2. : natural exponential function; 3. : summation from j=1 to n (n = number of genes); 4. : coefficient (log-hazard ratio) for gene j estimated by the Cox model; 5. : relative expression level of gene j (e.g., log2-transformed, standardized); 6. : patient-level prognostic index computed from the model; 7. : follow-up time;
Comments 12: Results section: (i) Figures have very small fonts, making them difficult or even impossible to read. Response 12: We thank the reviewer for pointing this out and fully agree. We have carefully reformatted all figures and increased font sizes for axis labels, legends, and annotations to ensure readability at the journal’s print scale. We also standardized line thickness, added clear scale bars where appropriate, and checked that all labels remain legible when reduced to one- or two-column width. These adjustments have been applied throughout the Results figures (Figs. 1–7).
Comments 13: (iii) The result descriptions repeat what is seen in the graphs rather than interpreting them. For example: "Tumor volume was significantly smaller in the HPX-overexpressed group (Fig. 7B)" – it would be appropriate to immediately add "indicating that HPX inhibits tumor growth in vivo." Response 13: We thank the reviewer for these valuable suggestions and fully agree. We have revised the Results to add immediate interpretive statements. Below are the direct replacements: 1. Tumors derived from HPX-overexpressed cells were significantly smaller in size com-pared to controls, as shown by representative images, indicating that HPX inhibits tumor growth in vivo. (Page 17, line 463-465) 2. HPX-overexpressed cells exhibited significantly higher proportions of apoptotic cells in Bel-7402, HCC-LM3, and MHCC-97L lines (Fig. 6A), supporting a pro-apoptotic role. (Page 14, line 437-439) 3. HPX-overexpressed cells showed delayed migration compared to controls in Bel-7402 (Fig. 5K), HCC-LM3 (Fig. 5L), and MHCC-97L (Fig. 5M) cells, indicating suppression of proliferative. (Page 13, line 421-423)
Comments 14: (iv) In the bioinformatics analysis (pp. 11–12): the description of the risk model and regression coefficients is too technical – it would be worthwhile to simplify it. Response 14: We thank the reviewer for this helpful suggestion and fully agree. We have streamlined the Results to focus on interpretation and moved technical details: 1. We randomly split the TCGA-LIHC cohort (follow-up > 0) into training and internal test sets (5:5). Fibrinolysis-related signature were first screened by univariable Cox analysis, then LASSO-Cox were performed in the training set (Fig. 3E–F). After multivariable Cox regression, six genes (ACAT1, GRHPR, HPX, PCK2, IYD, and PON1) were retained to construct the prognostic signature. For each patient, a risk signature was calculated as follow: . (Page 12, line 376-382)
Comments 15: Discussion section: It is worth highlighting a strength of the discussion: the context of fibrinolysis and the role of HPX is well presented. Notes: (i) In this section, in many places, the authors repeated the results from the Results section rather than interpreting them. Response 15: We sincerely thank the reviewer for highlighting this point and fully agree. We have rewritten the Discussion to emphasize interpretation, implications, and limitations rather than re-stating results. Below are the direct replacement paragraphs. 1. This study systematically investigated the role of fibrinolysis in pan-cancer, we successfully identified fibrinolysis-related gene HPX influencing HCC progression. Functional experiments demonstrated that HPX aligned with suppression of tumor phenotypes. Clinically, as a secreted protein, HPX has biomarker potential for risk as-sessment. These findings nominate HPX as a candidate for clinical stratification and therapeutic exploration, while underscoring the need for validation in larger, etiologi-cally diverse cohorts and for pathway-level perturbation studies to define causality. (Page 19, line 487-493)
Comments 16: (ii) There is no clear comparison with other studies that indicate HPX as a progression promoter in some cancers (e.g., PDAC – mentioned, but not elaborated upon). Response 16: We thank the reviewer and agree to provide a clearer cross-cancer comparison. We retained the pancreatic ductal adenocarcinoma (PDAC) study and clarified that other reports mainly detected HPX elevation in patient fluids without mechanistic follow-up. We have replaced the prior generic statement in the Discussion with the text below: 1. HPX has been linked to disease progression in pancreatic ductal adenocarcinoma (PDAC): stromal HPX expression correlated with lymph-node metastasis, suggesting that HPX may serve as a biomarker or potential therapeutic target in PDAC. (Page 19, line 511-514)
Comments 17: (iii) Furthermore, I believe that this section lacks a clear emphasis on potential translational limitations, e.g., differences between mouse models and patients. These limitations are mentioned (p. 20), but are very general. Response 17: We thank the reviewer for this important point and fully agree. We have expanded the Discussion to specify concrete translational limitations and the steps we will take to address them. The following text has been added: 1. Third, our in-vivo evidence comes from xenografts in immunodeficient mice that lack cirrhosis; responses may differ in immunocompetent or fibrotic livers. Etiologic heter-ogeneity in HCC (HBV/HCV, alcohol-related, NASH) and rebalanced hemostasis can modify fibrinolysis programs and HPX biology. Because HPX is secreted, circulating levels may be influenced by liver function, hemolysis, inflammation, and preanalytical handling. (Page 20, line 554-559)
Comments 18: Substantive errors: (i) The abstract states that fibrinolysis "serves as a protective factor for HCC across pan-cancers." This wording is imprecise. It should be: "...is associated with protective outcomes in HCC compared to other cancers." Response 18: We thank the reviewer for this precise correction and fully agree. We have revised the abstract wording to avoid an overgeneralized “protective factor” claim and to state the association clearly. 1. HPX is associated with protective outcomes and a potential therapeutic target in HCC. (Page 19, line 509)
Comments 19: (ii) In my opinion, the HPX–TNF-α–Bax/Bcl-2 relationship is inferred primarily based on correlation and expression; however, strong causal data are lacking (e.g., no experiments with TNF-α inhibitors). Response 19: We thank the reviewer for this important point and fully agree. Our current data indicate associations consistent with TNF-α–linked mitochondrial apoptosis but do not establish causality. We have therefore tempered wording and emphasized the limitations as below: 1. HPX was associated with an anti-tumor phenotype, consistent with TNF-α–mediated mitochondrial apoptosis, including an increased Bax/Bcl-2 ratio. (Page 2, line 53-54) 2. First, while we identified HPX as a critical regulator of apoptosis in HCC, strong rela-tionship data between HPX and TNF-α–Bax/Bcl-2 are lacking. (Page 20, line 550-552) 3. Importantly, HPX was associated with a tumor-suppressive phenotype and increased apoptosis, consistent with TNF-α–mediated mitochondrial signaling. (Page 20, line 564-566)
Comments 20: (iii) In the in vivo analyses (pp. 18–19), there is no clear information on the number of animals in each group; it is difficult to assess statistical reliability. After considering all my comments and suggestions, and after making corrections and additions, the manuscript may be reconsidered for submission, provided that the other reviewers also agree. Status: major revision. Response 20: We thank the reviewer for this important point and fully agree. We now state the exact number of animals per group in the Methods and directly in the figure legends for all in-vivo experiments. Randomization, blinding, and endpoints are also clarified. 1. For tumor implantation, 5×106 cells suspended in 120μL phosphate-buffered saline (PBS) were subcutaneously injected into the flank of each mouse (each group: n = 5). (Page 4, line 143-145) 2. Figure 7. HPX inhibited tumor growth and promotes apoptosis in vivo. (A) Tumors derived from MHCC-97L cells transfected with plvx-HPX (n = 5) or NC (n = 5). (Page 18, line 478-479)
|
|
4. Comments on the Quality of English Language |
|
Point 1: Style and language: the manuscript requires very careful editing by a native speaker well-versed in the technical vocabulary used in the manuscript. (i) The text contains numerous grammatical and stylistic errors, e.g., "Further research is needed essential to elucidate..." (should be: "Further research is essential to elucidate..."), "HPX could upregulated of Bax/Bcl-2..." (should be "HPX could upregulate Bax/Bcl-2..."), method cervical dislocation under anesthesia", etc. |
|
Response 1: We appreciate this comment and agree. However. We did not find the sentence “Further research is needed essential to elucidate …” in the current version; The phrasing issues noted by the reviewer have been addressed wherever they occurred across the Abstract, Methods, Results, and Discussion. |
|
Point 2: (ii) Please pay close attention to the tenses, e.g., some methods are described in the past tense, others in the present tense. |
|
Response 2: We thank the reviewer and fully agree. We conducted a line-by-line edit to standardize the entire Methods section to simple past tense, ensuring consistent usage across all subsections and figure legends. These revisions harmonize verb forms (“was/were performed,” “were obtained,” “were analyzed”), resolve subject–verb agreement, and eliminate mixed tenses, while leaving scientific content unchanged. |
|
Point 3: (ii) Please explain all abbreviations where they are used for the first time (e.g., "FpA" appears without prior explanation in the main text, only in the abstract). |
|
Response 3: We thank the reviewer and fully agree. We audited the manuscript and now define every abbreviation at its first occurrence in the main text and figure legends, and then use the abbreviation consistently thereafter. We also minimized non-essential abbreviations. 1. Basically, fibrinogen is cleaved by thrombin to release fibrinopeptides A and B (FpA/FpB), exposing polymerization sites that drive fibrin assembly. (Page 2, line 69-70) |
Reviewer 2 Report
Comments and Suggestions for Authors
The manuscript presents a comprehensive and well-structured analysis exploring the role of fibrinolysis in hepatocellular carcinoma (HCC), with a particular focus on hemopexin (HPX). The integration of pan-cancer bioinformatics, molecular subtyping, and experimental validation is a significant strength, as it demonstrates a consistent and biologically plausible tumor-suppressive role for HPX through TNF-α–mediated apoptosis. The study is methodologically rigorous, and the combination of in vitro and in vivo findings provides strong support for the conclusions drawn. The clarity of figures and the logical flow from bioinformatics discovery to experimental validation further enhance the impact of the work.
That said, there are areas where the manuscript could be further strengthened. While the study convincingly demonstrates the association between HPX and apoptotic pathways, the mechanistic exploration remains somewhat limited. Additional functional experiments or rescue assays could better delineate whether the observed pro-apoptotic effect is a direct consequence of HPX activity or mediated through upstream/downstream interactions. Furthermore, while the risk signature appears robust, additional validation in independent, larger, and more diverse cohorts would increase confidence in its clinical applicability. A more detailed discussion of potential translational implications—such as the feasibility of HPX as a biomarker in clinical practice or as a therapeutic target—would also improve the manuscript’s relevance to both researchers and clinicians.
Overall, this is a valuable contribution to the field of HCC research. With some refinement in the mechanistic depth and expanded validation of the prognostic signature, the study could have strong clinical and translational implications.
Author Response
|
Comments 1: The manuscript presents a comprehensive and well-structured analysis exploring the role of fibrinolysis in hepatocellular carcinoma (HCC), with a particular focus on hemopexin (HPX). The integration of pan-cancer bioinformatics, molecular subtyping, and experimental validation is a significant strength, as it demonstrates a consistent and biologically plausible tumor-suppressive role for HPX through TNF-α–mediated apoptosis. The study is methodologically rigorous, and the combination of in vitro and in vivo findings provides strong support for the conclusions drawn. The clarity of figures and the logical flow from bioinformatics discovery to experimental validation further enhance the impact of the work. That said, there are areas where the manuscript could be further strengthened. While the study convincingly demonstrates the association between HPX and apoptotic pathways, the mechanistic exploration remains somewhat limited. Additional functional experiments or rescue assays could better delineate whether the observed pro-apoptotic effect is a direct consequence of HPX activity or mediated through upstream/downstream interactions. Furthermore, while the risk signature appears robust, additional validation in independent, larger, and more diverse cohorts would increase confidence in its clinical applicability. A more detailed discussion of potential translational implications—such as the feasibility of HPX as a biomarker in clinical practice or as a therapeutic target—would also improve the manuscript’s relevance to both researchers and clinicians. Overall, this is a valuable contribution to the field of HCC research. With some refinement in the mechanistic depth and expanded validation of the prognostic signature, the study could have strong clinical and translational implications. |
|
Response 1: We sincerely thank the reviewer for the positive assessment and thoughtful suggestions. In light of the request to avoid overstatement, we have tempered the causal wording throughout (Abstract/Results/Discussion) so that conclusions are framed as associations consistent with tumor necrosis factor-α (TNF-α)–linked mitochondrial apoptosis, rather than definitive causality. Regarding clinical translation, we have expanded the Discussion to address biomarker feasibility and therapeutic implications as requested. we also emphasized the need for larger, diverse external cohorts to reinforce the prognostic signature’s generalizability. 1. Many Asian patients with HCC have underlying chronic hepatitis B or C infection and cirrhosis, experience ‘rebalanced’ hemostasis due to impaired hepatic synthesis, portal hypertension–related thrombocytopenia, and inflammation. These changes can shift coagulation–fibrinolysis dynamics and confound circulating markers, including HPX. Accordingly, clinical translation of HPX and fibrinolysis-based signatures should account for liver function, etiology (HBV/HCV, alcohol-related, NASH), and anticoagulant exposure. Future validation will stratify analyses by these factors and employ standardized preanalytical handling to ensure interpretability across diverse patient populations. To strengthen clinical applicability, multi-center validation with prespecified endpoints in larger and etiologically diverse cohorts is needed. (Page 20, line 540-549) |
Reviewer 3 Report
Comments and Suggestions for Authors
The article with the title “Pan-cancer analysis identifies hemopexin as a tumor suppressor in HCC via TNF-alpha-mediated mitochondrial apoptosis” is in generally well done, but I would offer these comments to the investigators:
- Some minor grammatical errors occur. The manuscript contains significant language-related issues. Please correct these types of grammatical errors throughout the paper.
- Introduction lines 58-59: Since the primary focus of the paper is liver cancer, I strongly recommend discussing the main causes of hepatocellular carcinoma (HCC), such as hepatitis virus infection and liver fibrosis.
- Results: "I recommend performing an experiment to identify the subcellular localization of HPX in these cell lines."
- Results: Identifying these results in samples from patients with different cancer types would be a valuable addition.
- Discussion: I strongly recommend discussing in depth the role of low-molecular-weight heparin, which is widely used in cancer management
- Discussion: The majority of patients with liver metastases or hepatocellular carcinoma (HCC) may develop liver failure, a condition that affects coagulation and subsequently the fibrinolytic process. I recommend that this issue be discussed in the manuscript.
- Please update the reference list, as the majority of the cited studies are 10–20 years old.
Author Response
|
Comments 1: The article with the title “Pan-cancer analysis identifies hemopexin as a tumor suppressor in HCC via TNF-alpha-mediated mitochondrial apoptosis” is in generally well done, but I would offer these comments to the investigators: Some minor grammatical errors occur. The manuscript contains significant language-related issues. Please correct these types of grammatical errors throughout the paper. Introduction lines 58-59: Since the primary focus of the paper is liver cancer, I strongly recommend discussing the main causes of hepatocellular carcinoma (HCC), such as hepatitis virus infection and liver fibrosis. |
|
Response 1: We thank the reviewer for the helpful comments and agree. We have performed a thorough language edit across the manuscript and standardized usage and abbreviations. We also added a concise background paragraph on the main causes of hepatocellular carcinoma (HCC) at the start of the Introduction section. The new text is below: 1. Hepatocellular carcinoma (HCC) most commonly develops on a background of chronic liver disease. Major causes include chronic infection with hepatitis B virus (HBV) or hepatitis C virus (HCV) and long-standing hepatic fibrosis or cirrhosis arising from alcohol-related liver disease or non-alcoholic steatohepatitis (NASH) [3]. (Page 2, line 61-65)
Reference [3]. Singal AG; Llovet JM; Yarchoan M; et al. AASLD Practice Guidance on Prevention, Diagnosis, and Treatment of Hepatocellular Carcinoma. Hepatology. 2023;78(6):1922–1965. doi:10.1097/HEP.0000000000000466.
|
|
Comments 2: Results: "I recommend performing an experiment to identify the subcellular localization of HPX in these cell lines." |
|
Response 2: We thank the reviewer for this helpful suggestion and agree that documenting localization strengthens the study. Because hemopexin (HPX) is a secreted protein, subcellular immunostaining in cultured cells can be of limited interpretive value. We therefore assessed localization in xenograft tissues by immunohistochemistry and added the result to Fig. 7F as shown below. HPX staining showed a predominantly pericellular/extracellular pattern consistent with a secreted protein, and the intensity was higher in HPX-overexpressing tumors than in controls.
|
|
Comments 3: Results: Identifying these results in samples from patients with different cancer types would be a valuable addition. |
|
Response 3: We thank the reviewer for this valuable suggestion and agree. As summarized in Fig. 4D: HPX expression was highest in liver and bile duct, which is consistent with HPA and GTEx data as shown below. And, at the tumor level, HCC and cholangiocarcinoma showed lower HPX expression than adjacent normal tissues. These cross-cancer observations reinforce the tissue specificity of HPX.
|
|
Comments 4: Discussion: I strongly recommend discussing in depth the role of low-molecular-weight heparin, which is widely used in cancer management |
|
Response 4: We thank the reviewer and agree to add this clinical context. We have inserted the concise paragraph below into the Discussion. 1. Biologically, fibrinolysis can modulate tumor progression through extracellular matrix remodeling, angiogenesis, and metastatic dynamics; clinical practice also underscores this axis, as low-molecular-weight heparin (LMWH) is widely used for cancer-associated thrombosis and may influence the coagulation–fibrinolysis balance [38]. Preclinical microsatellite stable (MSS) colorectal cancer models demonstrate that LMWH can synergize with adoptive cell transfer or anti-PD-1, improving tumor vascular normalization and CD8⁺ T-cell infiltration to suppress growth and liver metastasis [39]. (Page 19, line 496-503) Reference [38]. Lee AYY, Levine MN, Baker RI, Bowden C, Kakkar AK, Prins M, Rickles FR, Julian JA, Haley S, Kovacs MJ et al. Low-molecular-weight heparin versus a coumarin for the prevention of recurrent venous thromboembolism in patients with cancer. N Engl J Med.2003; 349(2):146-153 [39]. Quan Y, He J, Zou Q, Zhang L, Sun Q, Huang H, Li W, Xie K, Wei F. Low molecular weight heparin synergistically enhances the efficacy of adoptive and anti-PD-1-based immunotherapy by increasing lymphocyte infiltration in colorectal cancer. J Immunother Cancer.2023; 11(8). |
|
Comments 5: Discussion: The majority of patients with liver metastases or hepatocellular carcinoma (HCC) may develop liver failure, a condition that affects coagulation and subsequently the fibrinolytic process. I recommend that this issue be discussed in the manuscript. Response 5: We thank the reviewer and agree. We have added a concise discussion highlighting how liver failure and cirrhosis reshape hemostasis and fibrinolysis in in Asian HCC patients, and how this affects interpretation and translation of our findings. 1. Many Asian patients with HCC have underlying chronic hepatitis B or C infection and cirrhosis, experience ‘rebalanced’ hemostasis due to impaired hepatic synthesis, portal hypertension–related thrombocytopenia, and inflammation. These changes can shift coagulation–fibrinolysis dynamics and confound circulating markers, including HPX. Accordingly, clinical translation of HPX and fibrinolysis-based signatures should account for liver function, etiology (HBV/HCV, alcohol-related, NASH), and anticoagulant exposure. Future validation will stratify analyses by these factors and employ standardized preanalytical handling to ensure interpretability across diverse patient populations. To strengthen clinical applicability, multi-center validation with prespecified endpoints in larger and etiologically diverse cohorts is needed. (Page 20, line 540-549)
|
|
Comments 6: Please update the reference list, as the majority of the cited studies are 10–20 years old. |
|
Response 6: We thank the reviewer and agree. We audited the bibliography and replaced older citations with recent primary studies and reviews across all key topics. We removed the outdated references #11, #14, and #18 and renumbered the in-text citations accordingly.
|